⊖ | **Open Peer Review** | Antimicrobial Chemotherapy | Research Article

# Molecular characterization of carbapenemase-producing Enterobacterales in a tertiary hospital in Lima, Peru

Diego Cuicapuza,[1,2,3,4] Steev Loyola,[1,2,5] Jorge Velásquez,[6] Nathaly Fernández,[6] Carlos Llanos,[2] Joaquim Ruiz,[7] Pablo Tsukayama,[3,4,8,9] Jesús Tamariz[1,2]

**ABSTRACT**    Carbapenemase-producing Enterobacterales (CPE) are a growing threat to global health and the economy. Understanding the interactions between resistance and virulence mechanisms of CPE is crucial for managing difficult-to-treat infections and informing outbreak prevention and control programs. Here, we report the characterization of 21 consecutive, unique clinical isolates of CPE collected in 2018 at a tertiary hospital in Lima, Peru. Isolates were characterized by phenotypic antimicrobial susceptibility testing and whole-genome sequencing to identify resistance determinants and virulence factors. Seven *Klebsiella pneumoniae* isolates were classified as extensively drug-resistant. The remaining *Klebsiella*, *Enterobacter hormaechei,* and *Escherichia coli* isolates were multidrug-resistant. Eighteen strains carried the metallo-β-lactamase NDM-1, two the serine-carbapenemase KPC-2, and one isolate had both carbapenemases. The $bla_{NDM-1}$ gene was located in the truncated ΔISAba125 element, and the $bla_{KPC-2}$ gene was in the Tn4401a transposon. ST147 was the most frequent sequence type among *K. pneumoniae* isolates. Our findings highlight the urgent need to address the emergence of CPE and strengthen control measures and antibiotic stewardship programs in low- and middle-income settings.

**IMPORTANCE**    Genomic surveillance of antimicrobial resistance contributes to monitoring the spread of resistance and informs treatment and prevention strategies. We characterized 21 carbapenemase-producing Enterobacterales collected at a Peruvian tertiary hospital in 2018, which exhibited very high levels of resistance and carried numerous resistance genes. We detected the coexistence of carbapenemase-encoding genes ($bla_{NDM-1}$ and $bla_{KPC-2}$) in a *Klebsiella pneumoniae* isolate that also had the PmrB(R256G) mutation associated with colistin resistance. The $bla_{KPC-2}$ genes were located in Tn4401a transposons, while the $bla_{NDM-1}$ genes were in the genetic structure Tn125 (ΔISAba125). The presence of high-risk clones among *Klebsiella pneumoniae* (ST11 and ST147) and *Escherichia coli* (ST410) isolates is also reported. The study reveals the emergence of highly resistant bacteria in a Peruvian hospital, which could compromise the effectiveness of current treatments and control.

**KEYWORDS**    carbapenems, carbapenem-producing Enterobacterales, drug resistance, Whole-genome sequencing, Peru

A ntimicrobial resistance (AMR) poses a significant threat to human health and the global economy. The rise of multidrug-resistant (MDR) and pan-resistant (PDR) bacteria jeopardizes the advancements in modern medicine, potentially leading us toward a post-antibiotic era (1). Carbapenems, vital antibiotics often reserved for severe infections caused by MDR Gram-negative bacilli, are now threatened by the emergence of carbapenemase-producing Enterobacterales (CPE) (2). These pathogens are notably

Address correspondence to Pablo Tsukayama, pablo.tsukayama@upch.pe.

The authors declare no conflict of interest.

See the funding table on p. 9.

difficult to treat due to the limited effectiveness of available drugs, forcing clinicians to revert to older medications such as colistin (2, 3).

Carbapenemases, as categorized by Ambler's classification, include Class A *Klebsiella pneumoniae* carbapenemase (KPC), Class B metallo-β-lactamases (MBLs) like New Delhi MBL (NDM), and Class D oxacillinases (OXA) such as OXA-48-like enzymes found in Enterobacterales (2). *Klebsiella pneumoniae* and *Escherichia coli*, among the CPE, have emerged as significant global health threats due to their high mortality rates (4). The widespread dissemination of CPE is primarily attributed to the horizontal transfer of antibiotic-resistance genes via mobile genetic elements, including plasmids and transposons (5). Various virulence factors enhance the pathogenicity of CPE. While there is extensive knowledge about the interplay between these factors and AMR in many bacterial pathogens (6), the specific genetic determinants that drive virulence and their interaction with resistance mechanisms are not fully understood (7).

In Peru, the first CPE infections were identified in late 2013 as a carbapenem-resistant *K. pneumoniae*, sequence type (ST) 340 of the high-risk clonal complex (CC) 258 (8, 9). Subsequent reports indicate the spread of various carbapenemases in clinical settings and non-traditional hosts such as *Providencia stuartii* (10, 11). By 2020, the spread of the metallo-β-lactamase NDM-1 in *K. pneumoniae* was evident, involving newer lineages like ST348 and ST147 (12). Despite numerous CPE reports in Peru and Latin America, comprehensive genomic data are needed to conduct in-depth evolutionary, molecular, and epidemiological studies. Understanding the relationships between different CPE strains, both locally and globally, is essential. To address this gap, we sequenced the genomes and analyzed the resistance and virulence determinants of 21 CPE samples from a tertiary hospital in Lima collected in 2018.

## RESULTS

### Antimicrobial susceptibility profiles

The results of antibiotic susceptibility testing of 12 *K. pneumoniae*, 1 *Klebsiella quasipneumoniae*, 2 *Klebsiella aerogenes*, 2 *E. coli*, and 4 *Enterobacter hormaechei* are shown in Fig. 1. Overall, all CPE isolates were resistant to penicillins plus β-lactamase inhibitors, cephalosporins, monobactams, and fluoroquinolones. Furthermore, nearly all CPE isolates were resistant to phenicols (19/21, 90.5%) and folate pathway inhibitors (18/21, 85.7%). All *Klebsiella* species were resistant to gentamicin, and almost all were susceptible to amikacin, whereas all *E. hormaechei* were resistant to amikacin but mainly susceptible to gentamicin. *E. coli* isolates were predominantly susceptible to both aminoglycosides. Resistance to fosfomycin (10/12, 83.3%) and colistin (10/12, 83.3%) was observed only among *K. pneumoniae* isolates, whereas the other CPE were susceptible to both drugs. Phenotypic screening of carbapenemase production resulted in the identification of 19 metallo-β-lactamase-producers and 3 serine-carbapenemase producers. Notably, strain KpCR02 tested positive for serine and metallo-carbapenemase enzymes using phenotypic methods. Seven *K. pneumoniae* strains were classified as extensively drug-resistant (XDR), and the remaining CPE as MDR (Fig. 1; Table S1).

### Genomic diversity

Four STs were identified among *K. pneumoniae* isolates: ST11 ($n = 1$), ST147 ($n = 8$), ST348 ($n = 1$), and ST4872 ($n = 2$) (Fig. 2). Among the *K. pneumoniae* ST147 genomes, they diverged by 21–115 single nucleotide polymorphisms (SNP). The *K. quasipneumoniae* isolate was classified as ST4952, and both *K. aerogenes* isolates were ST215. ST88 and ST471 were identified among *E. coli* isolates. The four *E. hormaechei* isolates were classified as ST2054.

### Antimicrobial resistance genes

*K. pneumoniae* isolates harbored more AMR genes (average = 16.8, SD = 3.3) than other species in this study (average = 14.0, SD = 3.4). The *bla*$_{OXA-1}$ gene was detected in

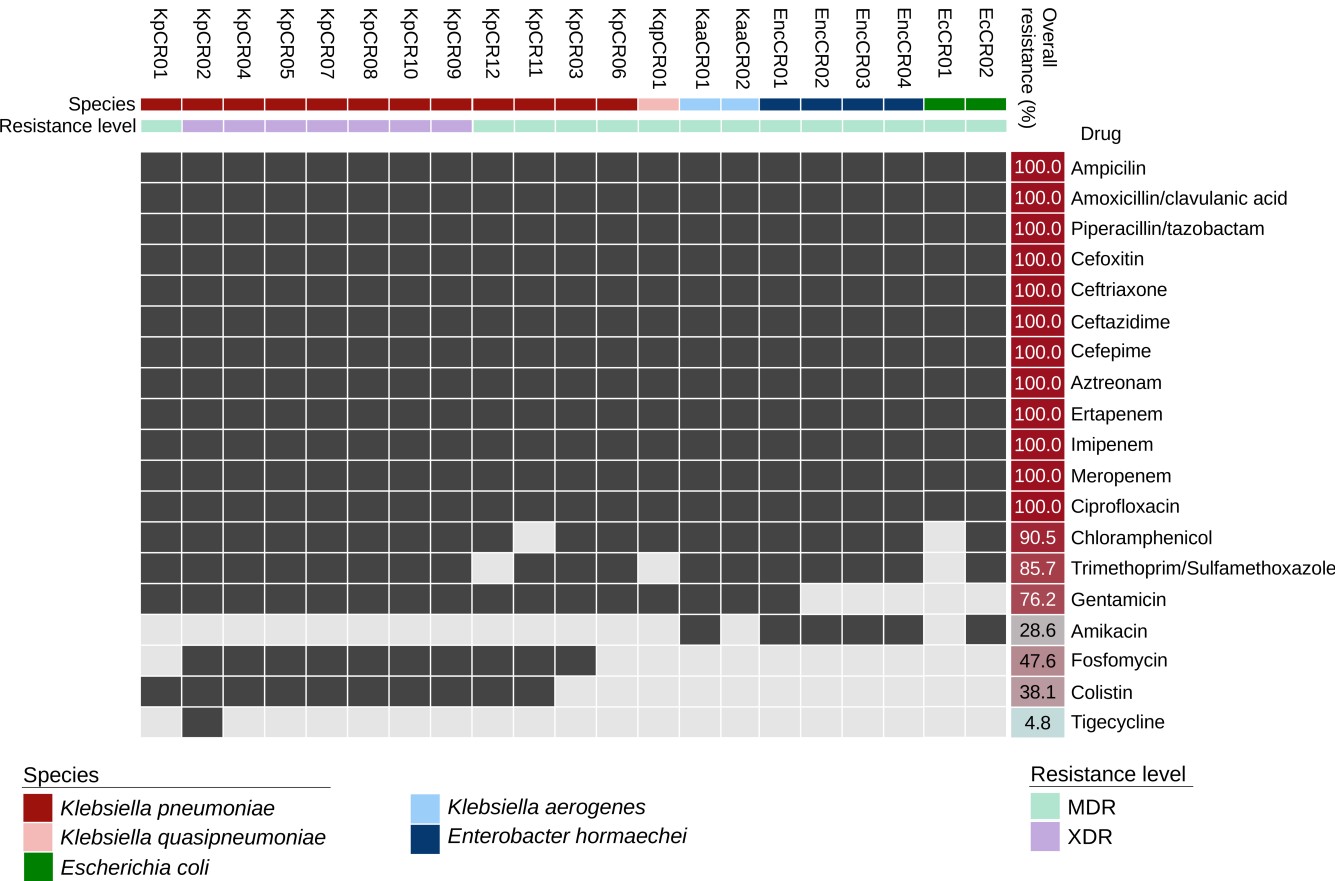

**FIG 1** Antimicrobial susceptibility profiles of Enterobacterales by antimicrobial category. Black and gray boxes indicate resistance and susceptibility, respectively.

all CPE isolates, while *sul1* and *arr*-3 (20/21, 95.2%), *catB3* (19/21, 90.5%), and *aac(6′)-lb-cr5* (18/21, 85.7%) genes were detected in nearly all isolates. Several AMR genes responsible for resistance to non-carbapenem β-lactams, including AmpC β-lactamases, extended-spectrum β-lactamases (ESBLs), aminoglycosides, fluoroquinolones, and other antimicrobial classes, were also detected among CPE isolates. Regarding carbapenemase-encoding genes, *bla*NDM-1 (19/21, 90.5%) was detected in almost all isolates, whereas *bla*KPC-2 (3/21, 14.3%) was detected in two *K. pneumoniae* and one *E. coli*. The *K. pneumoniae* strain KpCR02 tested positive for both *bla*NDM-1 and *bla*KPC-2 genes.

## Genetic context of carbapenemase genes

The *bla*KPC-2 gene was found in *K. pneumoniae* KpCR01 and KpCR02, and *E. coli* EcCR01, located in a Tn4401a transposon of the Tn3 family. The gene was flanked by ISKpn6 and ISKpn7 elements near the *tnpA* and *tnpR* transposase genes (Fig. 3A). The *bla*NDM-1 gene in *K. pneumoniae* strain KpCR09, *E. hormaechei* strain EnCCR02, and *E. coli* strain EcCR02 was found within the ΔTn125 element (Fig. 3B). The *bla*NDM-1 gene was flanked by ISA-ba125 and the *blaMBL* gene (Fig. 3B) in both reference strains and the three isolates described here. In *K. pneumoniae* strain KpCR09, the *blaTEM-26* and *aac (3)-lle* genes, which encode resistance to third- and fourth-generation cephalosporin and aminoglyco-side-modifying enzyme (AME), respectively, were identified upstream of ΔTn125.

## Virulence factors

*K. pneumoniae* isolate KpCR01 presented the yersiniabactin *ybt9* gene located on the integrative conjugative element (ICE) Kp3, along with the genes encoding for the *wzi50* capsular type and antigenic determinants K15 and O4. In contrast, isolate KpCR11

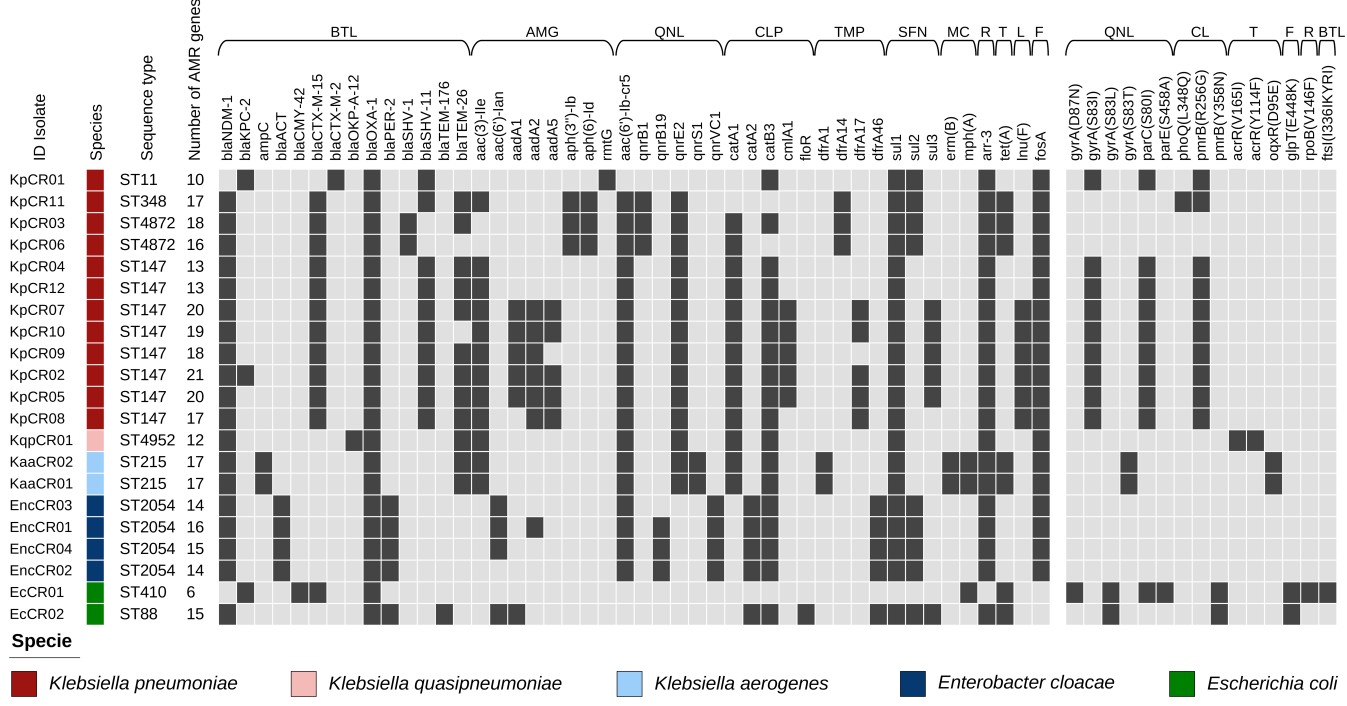

**FIG 2** Resistance genes of carbapenem-resistant Enterobacterales. Total AMR genes are summarized by bacterial isolate, and individual annotated genes conferring resistance are also summarized by antimicrobial category. BTL, β-lactams; AMG, aminoglycoside; QNL, quinolone; CLP, chloramphenicol; TMP, trimethoprim; SFN, sulfonamide; MC, macrolide; R, rifamycin; T, tetracycline; L, lincosamide; CL, colistin; and F, florfenicol.

exhibited the *ybt0* gene within the ICEKp12 element, accompanied by the *wzi94* capsular type gene and antigenic determinants K62 and O1. Notably, all *K. pneumoniae* ST147 isolates in our study had the aerobactin (*iuc5*) gene, genes encoding for the *wzi64* capsular type, and antigenic determinants KL64 and O2a. Additionally, in our two *E. coli* isolates, we identified several virulence genes, including *fimH*, associated with type 1 fimbriae; *iucC*, related to aerobactin synthetase; *iutA*, encoding the ferric aerobactin receptor; *lpfA*, indicative of long polar fimbriae; and *sitA*, associated with iron acquisition (Table S2).

## DISCUSSION

The rapid spread of CPE poses a significant threat to global public health. Our study of CPE from one Peruvian hospital in 2018 confirms the increasing prevalence of $bla_{NDM-1}$ over $bla_{KPC-2}$ in Peru since 2013, alongside the emergence of OXA-48-like carbapenemases, particularly OXA-181 (3–5, 13).

Our findings confirm high resistance levels in CPE, particularly in *Klebsiella pneumoniae*. Notably, the KpCR02 isolate exhibited up to 21 AMR genes, including NDM-1 and KPC-2. All NDM-1-producing CPE showed resistance to β-lactams, including aztreonam, often due to AmpC and ESBL genes, with the blaCTX-M-15 gene being particularly prevalent in our data set (14, 15). The co-production of carbapenemases NDM-1 and KPC-2 in *K. pneumoniae* has been previously described in South America, including Brazil, Argentina, Uruguay, Ecuador, and Paraguay (16). Our report provides further evidence of this occurrence and regional dispersion of CPE co-producing NDM-1 and KPC-2 carbapenemases.

The aminoglycoside resistance observed in our isolates could be attributed to genes that encode AMEs. The *rmtG* gene involved in 16S rRNA methylation confers high-level resistance to aminoglycosides (minimum inhibitory concentration, MIC ≥ 256 µg/mL) and is widely spread in *K. pneumoniae* found in South America (17). Among the AMEs, the acetyltransferases aac(3)-IIe and aac(6')Ib-cr5 were the most frequently detected. The

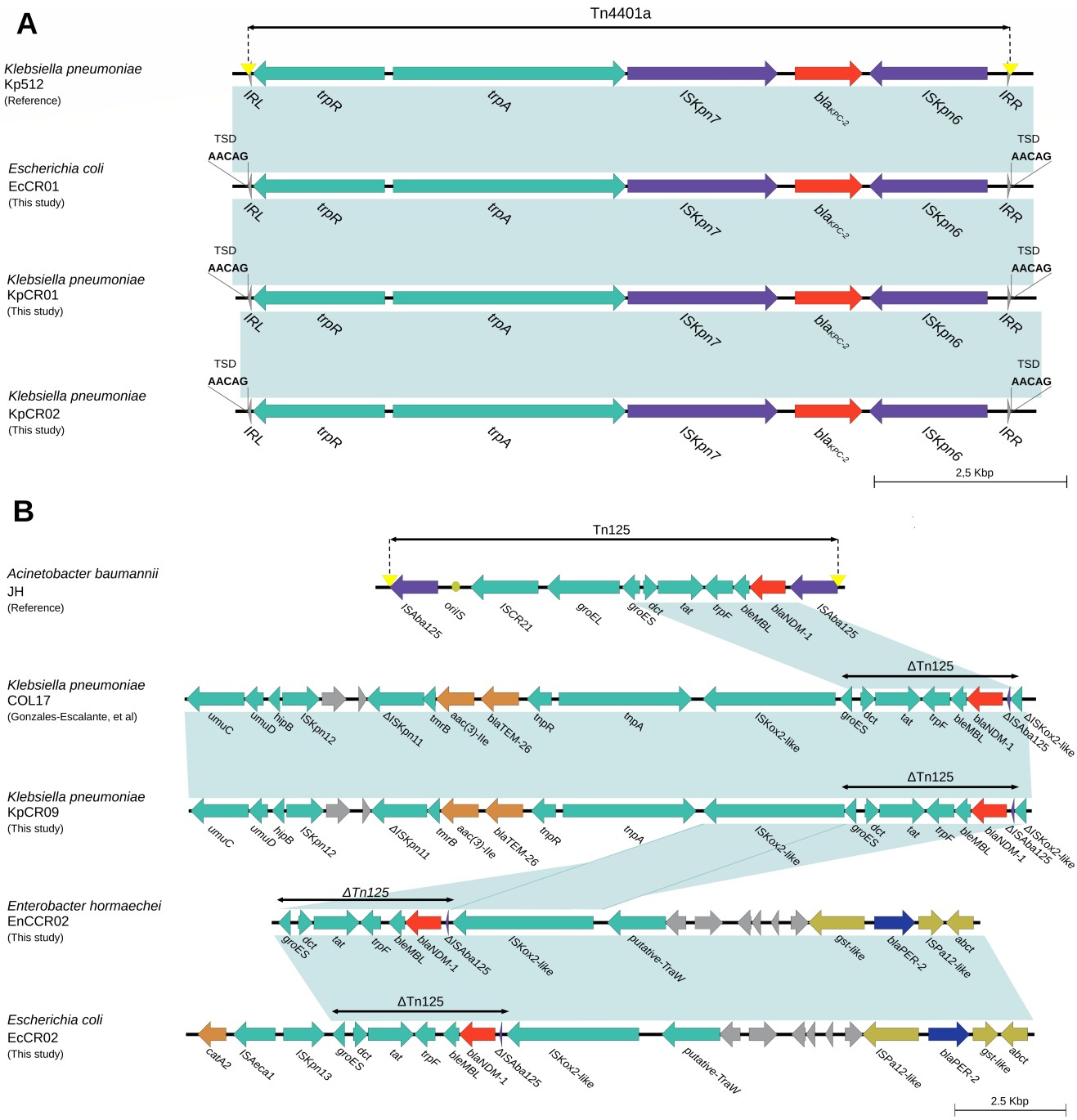

**FIG 3** (A) Genetic context of the structures of the Tn4401a transposon carrying $bla_{KPC-2}$ (GenBank accession number KT378596) and (B) genetic context of Tn125 (GenBank reference number JN872328) in our Enterobacterales isolates. Shading indicates 100% sequence similarity.

aac (3)-IIa has been associated with resistance to gentamicin, tobramycin, and netilmicin, as described elsewhere (18). The aac(6')Ib-cr is characterized by W102R and D179Y/G substitutions [both substitutions being present in aac(6')Ib-cr5], which, in addition to its capacity to inactivate several aminoglycosides, also result in the ability to inactivate several fluoroquinolones such as ciprofloxacin and norfloxacin (19). Nine *K. pneumoniae* and all *E. coli* isolates had amino acid substitutions at GyrA and ParC, the most common quinolone resistance mechanism (20). The remaining CPE did not present mutations in the established quinolone targets. However, all presented at least two transferable

mechanisms of quinolone resistance and alterations in efflux pump regulators in one case. While highly unusual several years ago, this scenario is increasingly being described (19, 20).

Resistance to colistin observed in *K. pneumoniae* isolates could be explained by the amino acid substitution detected in the PmrB (R256G) of 10 isolates, one of which also had the PhoQ(L348Q) mutation. Remarkably, both mutations are associated with lipid A modification (21). PmrB amino acid substitution (Y358N) was detected in two *E. coli* isolates; however, both were susceptible to colistin. Although the polymorphism may not be ruled out, this amino acid substitution has been previously described in *E. coli* isolates with high levels of colistin resistance (MIC >32 µg/mL) in the absence of *mcr* genes or other alteration in PmrAB, PhoPQ, or MgrB. Therefore, this single substitution may play a role in the development of colistin resistance (22).

Our genomic analysis found ST147 to be the dominant clone in *K. pneumoniae*, known for its high drug resistance and global dissemination. This includes the KpCR02 isolate, exhibiting both KPC-2 and NDM-1 and colistin resistance through pmrB mutations. The global emergence of KPC has been influenced by the wide dispersion of *K. pneumoniae* CC258, including ST11 and ST340 (23). ST11 is endemic in Brazil and several European countries (24, 25), and ST340 has been previously reported in Peru (7). Several *K. pneumoniae* ST11, ST147, and ST348 outbreaks have also been reported in Peru (10, 26). Our report of *K. pneumoniae* KPC-producing ST11 adds evidence of a high-risk international clone in Peru. In contrast to other reports, *K. pneumoniae* ST348 isolate KpCR11 described in this study significantly differs from other ST348 strains characterized by CTX-M-15 and KPC-3 production (27). The study reports *K. pneumoniae* ST4872, *K. quasipneumoniae* ST4952, and *K. aerogenes* ST215 for the first time in Peru. Among *E. coli* strains, ST471 and ST88 were identified, known for their resistance and virulence, alongside the first reported cases of *E. hormaechei* ST2054 in the region.

Analysis of the genetic context of the $bla_{KPC-2}$ gene revealed that it was located in the Tn4401a transposon, a member of the Tn3 transposon family (28, 29). To our knowledge, this is one of the few reports on Tn4401a in *E. coli* (30). Of note, current data differ from the previous genetic environment of $bla_{KPC-2}$ detected in Peru, highlighting the presence of different genetic structures carrying this gene in the country (7). Furthermore, the analysis of the genetic context of the $bla_{NDM-1}$ gene revealed that it was located in a truncated structure of Tn125. The architecture of the characterized ΔTn was identical to those previously reported in Peru and other Latin American countries (31, 32), and no differences were observed among the other CPE described here.

Hypervirulence is associated with the presence of additional siderophores, yersinia-bactin (*ybt*), aerobactin (*iuc*), salmochelin (*iro*), and specific capsular serotypes (K1, K2, and K5) (33). Interestingly, the aerobactin *iuc5* gene was detected in all *K. pneumoniae* ST147 samples. Besides potentially playing a critical role *in vitro* and *in vivo* virulence, it is a marker to identify highly virulent strains (33). This finding is compatible with that described in other CC258 isolates (34); however, in our isolate, neither characteristic virulence determinants such as salmochelin or colibactin nor genetic determinants associated with hypermucoviscosity (*rmpA*) were detected. To date, 12 distinct O loci have been identified, with O1 and O2 as the most common antigens (35), while K-type variation has been associated with six conserved genes (*galF, orf2, wzi, wza, wzb,* and *wzc*). The *wzi* gene best predicts K antigen-related virulence (36).

The study's limitations include a need for gene expression data and a limited sampling period from one healthcare center in 2018, which restricts the broader application of our findings. Instead, the results described here should be considered reference data and a starting point for further studies using more complex designs and sampling methods.

Overall, our comprehensive analysis of 21 CPE isolates underscores the significant challenge posed by drug-resistant CPE strains, emphasizing the need for expanded genomic surveillance, effective infection control, and antimicrobial stewardship

programs to curb the spread of CPE and maintain the efficacy of antimicrobial agents, particularly in the understudied Latin American region.

## MATERIALS AND METHODS

### Clinical isolates

From January to December 2018, 21 consecutive, non-replicated, and unique suspected CPE isolates were chosen based on their phenotypic resistance to meropenem at an MIC of ≥1 µg/mL using disk-diffusion assays. Clinical isolates were obtained from blood, urine, and other biological samples from inpatients and outpatients admitted to the tertiary public hospital, Hospital Nacional Arzobispo Loayza in Lima, Peru. The hospital's laboratory characterized all isolates at the species level using a standard biochemical test panel (37) and sent them to Universidad Peruana Cayetano Heredia (UPCH) for phenotypic and genomic characterization.

### Carbapenemase screening and identification

The phenotypic screening of carbapenemase production was done with the Triton Hodge Test and the carbapenem inactivation method as previously described (38). Inhibition tests using EDTA and phenylboronic acid were used to distinguish classes of carbapenemases (39).

### Antimicrobial susceptibility testing

Resistance phenotypes were determined using the VITEK2 System with AST-N249 cards (bioMérieux, France) following the manufacturer's instructions. MIC values were interpreted using clinical breakpoints established by the Clinical and Laboratory Standards Institute (CLSI) M100 guidelines (40). The European Committee for Antimicrobial Susceptibility Testing breakpoints were used for tigecycline interpretation since CLSI breakpoints were not available at the time of the study (41). Colistin susceptibility was assessed by the colistin broth disk elution test as previously described (42). Intermediate resistance results were analyzed and reported as resistant, and the criteria suggested by Magiorakos et al. were used to classify bacteria as MDR, XDR, or PDR (43).

### DNA extraction and whole-genome sequencing

Genomic DNA was extracted using the GeneJET DNA purification kit (Thermo Fisher Scientific, USA) from single colonies that were incubated in 1 mL of tryptic soy broth (Becton & Dickinson, USA) at 37°C and shaking at 300 rpm for 8 hours. The extracted gDNA was quantified with a Qubit 4.0 fluorometer, and Illumina libraries were constructed using 1 ng of gDNA with the Nextera XT kit (Illumina, USA). Paired-end libraries were sequenced on an Illumina MiSeq instrument generating 250-nt reads resulting in a mean per-base coverage of 75× (minimum 51× and maximum 108×).

### Bioinformatics analysis

Raw reads were assessed using FastQC v0.11.9 (http://www.bioinformatics.babraham.ac.uk/projects/fastqc/), trimmed with Trimmomatic v0.36.6 (LEADING:10, TRAILING:10, SLIDINGWINDOW:4:20, and MINLEN:40) (44), assembled *de novo* with SPAdes v3.10.0 (45), and evaluated for assembly quality using Quast v5.0.2 (46). Bacterial identification was performed using PathogenWatch (https://pathogen.watch/), and ST assignment was determined using the PubMLST (https://pubmlst.org/) scheme. Bacterial genomes with novel alleles were submitted to BIGSdb (https://pubmlst.org/software/bigsdb) for analysis and ST assignment. Virulence factors of *E. coli* were screened using VirulenceFinder v2.0 (https://cge.food.dtu.dk/services/VirulenceFinder/), and virulence factors and predicted capsule (K) and lipopolysaccharide (O) profiles of *Klebsiella* species were determined using Kleborate v2.3.2. AMR genes were identified using

AMRFinderPlus (coverage length ≥ 90%, nucleotide identity ≥ 90%, without gaps) (47). *E. coli* phylogroups were determined *in silico* using the ClermonTyping algorithm (http://clermontyping.iame-research.center/), and O and flagellar (H) serotypes were determined using SeroTypeFinder v2 (https://cge.food.dtu.dk/services/SerotypeFinder/).

## Genome comparisons

*K. pneumoniae* genomes belonging to ST147 were aligned to the reference genome NTUH-K2044, and recombinant regions were filtered out using Gubbins v3.3.0. Pairwise SNP differences between genomes were calculated and estimated using snp-dists (https://github.com/tseemann/snp-dists) (48). Contigs bearing the KPC-2 and NDM-1 genes were extracted from *de novo* assemblies and annotated with Prokka v1.5 (49). Since both genes were detected in transposable genetic elements, they were examined and annotated using MobileElementFinder v1.0.3 (https://pypi.org/project/MobileElementFinder/) and ISfinder (https://isfinder.biotoul.fr/) and then characterized using TETyper v1.1 (https://github.com/aesheppard/TETyper). Transposons were characterized using the Transposon Registry (http://transposon.lstmed.ac.uk/tn-registry) (50).

## ACKNOWLEDGMENTS

The study was funded by Prociencia grant number 088-2018. D.C. and P.T. are supported by a D43 TW007393 training grant awarded to UPCH by the Fogarty International Center of the U.S. National Institutes of Health. S.L. is supported by a competitive award received from Universidad de Cartagena–UNIMOL, as part of a Ph.D. scholarship.

## AUTHOR AFFILIATIONS

[1]Facultad de Medicina, Universidad Peruana Cayetano Heredia, Lima, Peru

[2]Laboratorio de Resistencia Antibiótica e Inmunopatología, Facultad de Medicina, Universidad Peruana Cayetano Heredia, Lima, Peru

[3]Laboratorio de Genómica Microbiana, Facultad de Ciencias e Ingeniería, Universidad Peruana Cayetano Heredia, Lima, Peru

[4]Emerge (Emerging Diseases and Climate Change Research Unit), Facultad de Salud Pública y Administración, Universidad Peruana Cayetano Heredia, Lima, Peru

[5]Grupo de Investigación UNIMOL, Facultad de Medicina, Universidad de Cartagena, Cartagena de Indias, Colombia

[6]Departamento de Patología Clínica y Banco de Sangre, Hospital Nacional Arzobispo Loayza, Lima, Peru

[7]Grupo de Investigación en Dinámicas y Epidemiología de la Resistencia a Antimicrobianos-"One Health", Universidad Científica de Sur, Lima, Peru

[8]Instituto de Medicina Tropical Alexander von Humboldt, Universidad Peruana Cayetano Heredia, Lima, Peru

[9]Parasites and Microbes Programme, Wellcome Sanger Institute, Hinxton, United Kingdom

## AUTHOR ORCIDs

Diego Cuicapuza 🆔 http://orcid.org/0000-0002-5735-4614
Steev Loyola 🆔 http://orcid.org/0000-0001-5455-2423
Joaquim Ruiz 🆔 http://orcid.org/0000-0002-4431-2036
Pablo Tsukayama 🆔 http://orcid.org/0000-0002-1669-2553
Jesús Tamariz 🆔 http://orcid.org/0000-0002-0827-8117

## FUNDING

| Funder | Grant(s) | Author(s) |
|---|---|---|
| CONCYTEC \| Fondo Nacional de Desarrollo Científico, Tecnológico y de Innovación Tecnológica (FONDECYT) | 088-2018 | Pablo Tsukayama |
| HHS \| NIH \| Fogarty International Center (FIC) | TW007393 | Diego Cuicapuza |
| | | Pablo Tsukayama |

## AUTHOR CONTRIBUTIONS

Diego Cuicapuza, Conceptualization, Data curation, Formal analysis, Investigation, Methodology, Validation, Visualization, Writing – original draft, Writing – review and editing | Steev Loyola, Data curation, Formal analysis, Investigation, Methodology, Supervision, Writing – original draft | Jorge Velásquez, Conceptualization, Methodology, Validation | Nathaly Fernández, Conceptualization, Methodology, Supervision, Validation | Carlos Llanos, Conceptualization, Investigation, Methodology | Joaquim Ruiz, Formal analysis, Methodology, Validation, Writing – original draft, Writing – review and editing | Pablo Tsukayama, Conceptualization, Data curation, Formal analysis, Funding acquisition, Methodology, Project administration, Resources, Supervision, Validation, Visualization, Writing – original draft, Writing – review and editing | Jesús Tamariz, Conceptualization, Data curation, Formal analysis, Investigation, Methodology, Supervision, Validation, Visualization, Writing – original draft, Writing – review and editing

## DATA AVAILABILITY

The raw read files and assemblies for the five isolates are available at NCBI under BioProject accession number PRJNA865026.

## ETHICS APPROVAL

The study protocol was approved by the Institutional Review Board of the Universidad Peruana Cayetano Heredia (SIDISI 100535) and approved by the Hospital Arzobispo Loayza research board. Patient information was anonymized and deidentified before analysis.

## ADDITIONAL FILES

The following material is available online.

### Supplemental Material

**Table S1 (Spectrum02503-23-s0001.xlsx).** MIC values for the studied isolates of carbapenemase-producing Enterobacterales (CPE).
**Table S2 (Spectrum02503-23-s0002.xlsx).** Virulence and antimicrobial resistance genes, along with assembly statistics of studied isolates of CPE.

### Open Peer Review

**PEER REVIEW HISTORY (review-history.pdf).** An accounting of the reviewer comments and feedback.

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
