## [Reviewer comments · Microbiology Spectrum]

Microbiology Spectrum

Molecular characterization of Carbapenemase-producing *Enterobacterales* in a tertiary hospital in Lima, Peru

Diego Cuicapuza, Steev Loyola, Jorge Velásquez, Nathaly Fernández, Carlos Llanos, Joaquim Ruiz, Pablo Tsukayama, and Jesus Tamariz

Corresponding Author(s): Pablo Tsukayama, Universidad Peruana Cayetano Heredia

Review Timeline:

Submission Date:	June 14, 2023
Editorial Decision:	July 12, 2023
Revision Received:	September 13, 2023
Editorial Decision:	October 1, 2023
Revision Received:	November 23, 2023
Accepted:	November 26, 2023

Editor: Cheryl Andam

Reviewer(s): Disclosure of reviewer identity is with reference to reviewer comments included in decision letter(s). The following individuals involved in review of your submission have agreed to reveal their identity: Dennis Nurjadi (Reviewer #1)

Transaction Report:

DOI: <https://doi.org/10.1128/spectrum.02503-23>

July 12, 2023

Dr. Pablo Tsukayama
Universidad Peruana Cayetano Heredia
Department of Microbiology
Av Honorio Delgado 430
Lima 15102
Peru

Re: Spectrum02503-23 (**Molecular characterization of Carbapenemase-producing *Enterobacterales* in a tertiary hospital in Lima, Peru**)

Dear Dr. Pablo Tsukayama:

Link Not Available

Sincerely,

Cheryl Andam

Journals Department
Reviewer comments:

Reviewer #1 (Comments for the Author):

The authors presented a study on CPE isolates from Peru (n=21). The study premise is of interest considering the global increase of AMR and the intercontinental spread of AMR. Overall the study is nicely written. However, some sections may benefit from shortening for a more focused and concise structure. Although mentioned in the discussion and methods section, no SNP data was presented in the manuscript. This would add some value to the findings.

Major comments

Lines 364-365: what was the rationale for using this definition for CPE (meropenem {greater than or equal to}1 mg/l)? This can be used to screen for CPE, but your study does not describe why this value is chosen and what happens after this selection. Maybe rephrase to "suspected CPE isolates were chosen based on".

Methods: how was species identification performed? What is considered "standard microbiological procedures"? MALDI-TOF MS?

- Methods: AST, which Vitek card was used?

- Methods/bioinformatics: QC parameters? Length trimming, minimum coverage, etc? Please add this information to the methods section.

- SNP distance: did you check for recombination sites using Gubbins? This should be done if not, as recombinations can overinflate the SNPs. Why is the SNP analysis not presented in the results section? It is interesting to see how the 8 ST11 K. pneumoniae isolates are related.

- Results/AMR genes: the text is redundant as the results are already presented in the Tables/Figures. Please focus on the major findings and maybe comment on the overall concordance of phenotypic genotypic, including colistin resistance.

- Line 197: genetic relatedness was not presented.. this should be added. I cannot find any information on this throughout the manuscript, but it was mentioned also in the methods section.

- Discussion: for the amount of data presented (only 21 sequenced isolates), the discussion is over proportionally long; please consider focusing on a few main findings and shortening the discussion for a more focused approach rather than artificially inflating the discussion with redundant results and figures. Please focus on putting your findings in context or other studies (epidemiologically) instead of listing the concordance of genotypic and phenotypic AMR. In general, you should focus on the molecular epidemiology in the discussion. AMR is already mentioned extensively in the results section.

- Lines 336-337: both are usually independent since the resistance gene is generally on different mobile genetic elements than the virulence factors, at least for hypervirulent Kp.

Minor comments

- Enterobacterales is an order and should not be written in italics.

- Why was the time period May to Dec 2018 chosen? Not Jan-Dec for example.

- Other limitation could be that the isolate collection was done in 2018. It has now been five years since, and the epidemiology and prevalence may have changed.

Reviewer #2 (Comments for the Author):

The authors suggest that the Magiorakos et al paper was used for MDR, XDR, PDR classifications. However, under this classification PDR isolates have to be non-susceptible to all agents tested. In this case the isolate in this study, KpCR01, would not be considered PDR. Further, isolates KpCR09, KpCR12, KpCR11, and KpCR03 would be considered MDR and not XDR. The text and analysis throughout the manuscript will need to be adjusted if the Magiorakos definition is used.

The analysis of virulence determinants is lacking. The authors screen for virulence factors of E. coli using VirulenceFinder and Klebsiella sp. using Kleborate and should provide this data in a supplementary table and discuss further in the results section. Were virulence determinants screened for in the Enterobacter isolates?

Line 63 - clarify "reported average". Where does that data come from?

Line 65 - Be specific about listing the chromosomal gene

Line 67 - This sentence makes it sound like this is the first report in E. coli. Consider re-writing for clarity.

Line 67- 68- abbreviate "Klebsiella" and "Escherichia"

Line 140 Antimicrobial Resistance Genes Results Section. The authors should discuss the presence of AmpC genes in the dataset as these are mentioned in the Discussion.

Line 289 - capitalize "The"

Line 298- "ST" should not be italicized

Line 322 - The manuscript does not address all of these virulence factors. Additional analysis is needed - LPS, K-types, fimbriae etc.

Line 421- Suggests only 5 isolates are available on NCBI but it looks like 21 SRAs have been uploaded.

Staff Comments:

Preparing Revision Guidelines

Please return the manuscript within 60 days; if you cannot complete the modification within this time period, please contact me. If you do not wish to modify the manuscript and prefer to submit it to another journal, please notify me of your decision immediately so that the manuscript may be formally withdrawn from consideration by Microbiology Spectrum.

RESPONSE TO REVIEWERS

We thank the reviewers for the thoughtful comments that have improved this manuscript. Below are point-by-point responses to the issues raised.

Reviewer #1

The authors presented a study on CPE isolates from Peru (n=21). The study premise is of interest considering the global increase of AMR and the intercontinental spread of AMR. Overall the study is nicely written. However, some sections may benefit from shortening for a more focused and concise structure. Although mentioned in the discussion and methods section, no SNP data was presented in the manuscript. This would add some value to the findings.

Thank you. We have revised all sections to improve flow and clarity and reduced the length of the introduction and discussion sections, as suggested. SNP analysis was included to further assess the relatedness of *K. pneumoniae* isolates belonging to the same ST (L105-106, L326-327). New text addressing the reviewer's comments have been highlighted in the marked-up document (yellow for reviewer 1's comments, green for reviewer 2's comments).

Major comments

1. Lines 364-365: what was the rationale for using this definition for CPE (meropenem {greater than or equal to}1 mg/l)? This can be used to screen for CPE, but your study does not describe why this value is chosen and what happens after this selection. Maybe rephrase to "suspected CPE isolates were chosen based on".
Comment addressed. The methods section was modified to indicate that these criteria were used to select isolates suspected of producing carbapenemases (L279-281).
2. Methods: how was species identification performed? What is considered "standard microbiological procedures"? MALDI-TOF MS?
We used a standard biochemical test panel to identify isolates based on the Bergey's Manual of Systematic Microbiology (L284-285).
3. Methods: AST, which Vitek card was used?
We used VITEK2 AST-N249 cards to determine resistance phenotypes (L293-205)
4. Methods/bioinformatics: QC parameters? Length trimming, minimum coverage, etc? Please add this information to the methods section.
We added the QC parameters for Trimmomatic (L312-313) and added the sequencing coverage calculations (L308-309).
5. SNP distance: did you check for recombination sites using Gubbins? This should be done if not, as recombinations can overinflate the SNPs. Why is the SNP analysis not presented in the results section? It is interesting to see how the 8 ST147 *K. pneumoniae* isolates are related.
We used Gubbins v3.3.0 to mask recombinant sites from our genome alignments (L327) and added SNP analysis of ST147 isolates to the results section (L105-106).
6. Results/AMR genes: the text is redundant as the results are already presented in the Tables/Figures. Please focus on the major findings and maybe comment on the overall concordance of phenotypic genotypic, including colistin resistance.

Comment addressed. We edited and shortened the results/AMR paragraph highlighting the most important results (L110-119).

7. Line 197: genetic relatedness was not presented.. this should be added. I cannot find any information on this throughout the manuscript, but it was mentioned also in the methods section.

SNP analysis was included to further assess the relatedness of *K. pneumoniae* isolates belonging to the same ST (L105-106, L326-327).

8. Discussion: for the amount of data presented (only 21 sequenced isolates), the discussion is over proportionally long; please consider focusing on a few main findings and shortening the discussion for a more focused approach rather than artificially inflating the discussion with redundant results and figures. Please focus on putting your findings in context or other studies (epidemiologically) instead of listing the concordance of genotypic and phenotypic AMR. In general, you should focus on the molecular epidemiology in the discussion. AMR is already mentioned extensively in the results section.

We edited and shortened the discussion section by focusing on a few main findings and the molecular epidemiology of AMR.

9. Lines 336-337: both are usually independent since the resistance gene is generally on different mobile genetic elements than the virulence factors, at least for hypervirulent Kp.

Comment addressed. To avoid confusion, text indicating the possibility of independent events was removed.

Minor comments

10. Enterobacterales is an order and should not be written in italics.

Comment addressed.

11. Why was the time period May to Dec 2018 chosen? Not Jan-Dec for example.

This was an error on our part. Thanks for spotting that. The correct study period was from January to December 2018 (L276).

12. Other limitation could be that the isolate collection was done in 2018. It has now been five years since, and the epidemiology and prevalence may have changed.

We have incorporated this additional limitation into the discussion (L259-L263).

Reviewer #2

1. The authors suggest that the Magiorakos et al paper was used for MDR, XDR, PDR classifications. However, under this classification PDR isolates have to be non-susceptible to

all agents tested. In this case the isolate in this study, KpCR01, would not be considered PDR. Further, isolates KpCR09, KpCR12, KpCR11, and KpCR03 would be considered MDR and not XDR. The text and analysis throughout the manuscript will need to be adjusted if the Magiorakos definition is used.

Comment addressed. The MDR, XDR and PDR classifications were adjusted according to the Magiorakos et al definition (L89-102, L296-298).

2. The analysis of virulence determinants is lacking. The authors screen for virulence factors of *E. coli* using VirulenceFinder and *Klebsiella* sp. using Kleborate and should provide this data in a supplementary table and discuss further in the results section. Were virulence determinants screened for in the *Enterobacter* isolates?

Comment addressed. We added a paragraph on virulence factors to the results section (L131-140) summarizing the data for *Klebsiella* by Kleborate and *E. coli* by VirulenceFinder available on the CGE website, where it has a curated base only for four bacterial species (*Listeria*, *S. aureus*, *E. coli* and *Enterococcus*), therefore the analysis of *Enterobacter* was not included.

3. Line 63 - clarify "reported average". Where does that data come from?

Comment addressed. The text was revised for improved clarity (L110-111).

4. Line 65 - Be specific about listing the chromosomal gene

Comment addressed.

5. Line 67 - This sentence makes it sound like this is the first report in *E. coli*. Consider re-writing for clarity.

Comment addressed.

6. Line 67- 68- abbreviate "*Klebsiella*" and "*Escherichia*"

Comment addressed.

7. Line 140 Antimicrobial Resistance Genes Results Section. The authors should discuss the presence of AmpC genes in the dataset as these are mentioned in the Discussion.

Comment addressed. Because we have few AmpC-producing isolates, the results are focused on more important genes such as NDM and KPC.

8. Line 289 - capitalize "The"

Comment addressed.

9. Line 298- "ST" should not be italicized

Comment addressed.

10. Line 322 - The manuscript does not address all of these virulence factors. Additional analysis is needed - LPS, K-types, fimbriae etc.

Comment addressed. Added K-types, O-types and other virulence factors provided by Kleborate (L131-140).

11. Line 421- Suggests only 5 isolates are available on NCBI but it looks like 21 SRAs have been uploaded.

Comment addressed. You can now find all the raw Illumina datasets and assemblies at the designated NCBI Bioproject (L338-339). See screenshot below.

Molecular characterization of Carbapenemase-producing Enterobacterales in a tertiary hospital in Lima, Peru

Accession: PRJNA865026 ID: 865026

This study presents the genomic analysis of the first clinical isolates of carbapenem-resistant Enterobacteriaceae (CRE) carrying blaNDM-1 and blaKPC-2 genes and their genetic environment from a health center in Lima, Peru.

Accession	PRJNA865026
Data Type	Genome sequencing and assembly
Scope	Multispecies
Submission	Registration date: 2-Aug-2022 Universidad Peruana Cayetano Heredia
Related Resources	Universidad Peruana Cayetano HerediaLaboratorio de Genómica MicrobianaLaboratorio de Resistencia a Antimicrobianos e Inmunopatología
Relevance	Medical

Project Data:

Resource Name	Number of Links
SEQUENCE DATA	
Nucleotide (WGS master)	21
SRA Experiments	21
OTHER DATASETS	
BioSample	21
Assembly	21

October 1, 2023

Dr. Pablo Tsukayama
Universidad Peruana Cayetano Heredia
Department of Microbiology
Av Honorio Delgado 430
Lima 15102
Peru

Re: Spectrum02503-23R1 (**Molecular characterization of Carbapenemase-producing *Enterobacterales* in a tertiary hospital in Lima, Peru**)

Dear Dr. Pablo Tsukayama:

Link Not Available

Sincerely,

Cheryl Andam

Journals Department
Reviewer comments:

Reviewer #1 (Comments for the Author):

Thank you for addressing the comments and revising your manuscript.

Reviewer #2 (Comments for the Author):

Failed to comprehensively address reviewer comments including, complete analysis of virulence factors found in the isolates with supplementary tables. Also, failed to provide analysis of AmpC genes in the *K. pneumoniae* isolates. The discussion talks at length about virulence factors and surface antigens but it is impossible for the reader to understand the virulence loci, KL, O type etc found in all the *K. pneumoniae* isolates (for example).

Staff Comments:

Preparing Revision Guidelines

Please return the manuscript within 60 days; if you cannot complete the modification within this time period, please contact me. If you do not wish to modify the manuscript and prefer to submit it to another journal, please notify me of your decision immediately so that the manuscript may be formally withdrawn from consideration by Microbiology Spectrum.

RESPONSE TO REVIEWERS (updated November 2023)

We thank the reviewers for a new round of comments that have improved this manuscript. Below are point-by-point responses to the issues raised.

Reviewer #1:

Please include supplementary file (table) that shows the genomic features (genome length, N50, number of contigs, other quality check results), AST results, ST, and presence/absence of resistance and virulence genes of individual isolates. Reviewer 2's comment about ampC was not addressed. Several comments by Reviewer were not addressed or were missing. When you address their comments, please write the lines where they are found instead of simply saying "Comment addressed". Lastly, the background information in the Introduction is sparse. Please expand the first two paragraphs to provide relevant context.

- Two supplementary tables are now included: the first table includes the results of antibiotic susceptibility testing, and the second table includes genomic characteristics (genome length, ST, and the presence/absence of resistance and virulence genes).
- We have revised and expanded the introduction section to provide more background information on CPE and mechanisms of carbapenem resistance (L54-71).

Reviewer #2:

Failed to comprehensively address reviewer comments including, complete analysis of virulence factors found in the isolates with supplementary tables. Also, failed to provide analysis of AmpC genes in the *K. pneumoniae* isolates. The discussion talks at length about virulence factors and surface antigens but it is impossible for the reader to understand the virulence loci, KL, O type etc found in all the *K. pneumoniae* isolates (for example).

- We have expanded the results section to include a paragraph on identified virulence factors (L122-131) and discuss these accordingly (L188-197).
- We also provide more details on AmpC beta-lactamases (L107, 137- 140). However, we kept it short because it was only found in two *E. coli* isolates, and we did not want to extend the manuscript further.
- We have also revised and streamlined the discussion section to improve clarity and flow. We hope you agree that the new version reads more easily than previous manuscript versions.

Re: Spectrum02503-23R2 (**Molecular characterization of Carbapenemase-producing *Enterobacteriales* in a tertiary hospital in Lima, Peru**)

Dear Dr. Pablo Tsukayama:

Your manuscript has been accepted, and I am forwarding it to the ASM production staff for publication. Your paper will first be checked to make sure all elements meet the technical requirements. ASM staff will contact you if anything needs to be revised before copyediting and production can begin. Otherwise, you will be notified when your proofs are ready to be viewed.

Sincerely,
Cheryl Andam
Editor
Microbiology Spectrum